# FIRST RETURN, ENTROPY-ELICITING EXPLORE

## ABSTRACT

Reinforcement Learning from Verifiable Rewards (RLVR) improves the reasoning abilities of Large Language Models (LLMs) but it struggles with unstable exploration. We propose **FR3E** (First Return, Entropy-Eliciting Explore), a structured exploration framework that identifies high-uncertainty decision points in reasoning trajectories and performs targeted rollouts to construct semantically grounded intermediate feedback. Our method provides targeted guidance without relying on dense supervision. Empirical results on mathematical reasoning benchmarks(AIME24) show that FR3E promotes more stable training, produces longer and more coherent responses, and increases the proportion of fully correct trajectories. These results highlight the framework's effectiveness in improving LLM reasoning through more robust and structured exploration.

## 1 INTRODUCTION

Reinforcement Learning (RL) significantly enhances the reasoning capabilities (Brown et al., 2020; Zhou et al., 2022; Wei et al., 2022; Ouyang et al., 2022; Wu et al., 2024; Forootani, 2025; Huan et al., 2025) of LLMs, particularly in complex tasks like mathematical problem-solving and code generation using RLVR (Shao et al., 2024; Yu et al., 2025; Yue et al., 2025; Guo et al., 2025). A central challenge in these RLVR methodologies is the granular assignment of credit to intermediate steps within a reasoning trajectory. Many approaches, such as Shao et al. (2024)'s Group Relative Policy Optimization (GRPO), use the final outcome reward to estimate the value of all intermediate actions. This simple method uniformly credits all steps, failing to distinguish pivotal reasoning from trivial actions, thereby constraining learning and diverging from the human intuition that steps vary in importance. For tasks with sparse and delayed rewards, this leads to imprecise credit assignment, hindering the model's ability to identify crucial actions and potentially causing issues like "overthinking" (Chen et al., 2025).

Existing strategies for estimating intermediate state or action values have notable limitations. Value model-based approaches like Schulman et al. (2017)'s Proximal Policy Optimization (PPO) and VAPO (Yue et al., 2025) employ a critic model. However, training a stable and accurate critic for the vast state space of LLMs is difficult, often leading to instability, bias, and significant computational overhead. Even with techniques like Generalized Advantage Estimation (GAE) (Schulman et al., 2018), reward propagation to distant tokens remains problematic. While GRPO avoids a critic, it primarily relies on trajectory-level rewards, not fully solving granular credit assignment. Other methods use sampling or heuristic-based intermediate reward estimation. For instance, VinePPO (Kazemnejad et al., 2024) segments trajectories heuristically and uses Monte Carlo (MC) sampling for step-level advantages. Process reward models (PRMs) (Luo et al., 2024; Lightman et al., 2023), as in S-GRPO (Dai et al., 2025) and PRIME (Cui et al., 2025a), aim to provide intermediate feedback but face challenges with heuristic quality, sampling variance, labeling costs for explicit PRMs, or the granularity of implicitly derived rewards. Fixed reward schemes may also lack adaptability. Overall, current methods suffer from complexity, instability, cost, and brittle heuristics.

In this work, we introduce **First Return, Entropy-Eliciting Explore** (FR3E), a structured exploration framework that identifies high-uncertainty decision points in reasoning trajectories and performs targeted rollouts to construct semantically grounded intermediate feedback. Unlike traditional methods that perform full rollouts starting directly from prompts, FR3E localizes high-entropy tokens along correctly completed reasoning trajectories as critical decision points. These tokens serve as anchors from which targeted rollouts are initiated, enabling structured exploration around uncertain yet pivotal steps in the reasoning process.

By performing partial rollouts from these key decision points, FR3E synthesizes localized feedback signals that would otherwise be unavailable in standard autoregressive generation. This process does not depend on existing reward signals or attempt to perform credit assignment over long sequences; instead, it actively constructs new sources of evaluation through controlled exploration. These synthesized signals are semantically grounded and particularly useful for guiding early-stage reasoning, where feedback is often most ambiguous.

The design of FR3E builds upon the principles of "First Return, Then Explore" (FRTE) (Ecoffet et al., 2021), adapted to the sequential generation behavior of LLMs. Compared to conventional reinforcement learning approaches that often treat all generation steps uniformly, FR3E leverages entropy profiles to identify critical reasoning junctures. It then performs targeted rollouts from these states, yielding structured exploration and more meaningful feedback, all without requiring detailed reward labels for every step.

We summarize our key contributions as follows:

1. **A Reliable Exploration Framework for Trajectory-Level Reward Shaping**: We introduce FR3E, a novel reinforcement learning algorithm that improves reward shaping at the trajectory level by emphasizing reliable exploration paths.

2. **Improved Training Stability and Reasoning Capability**: FR3E maintains entropy at a stable and gradually increasing level during training, preventing early collapse and enabling the model to generate longer and more reliable reasoning chains. This addresses common failures in standard RL, particularly in specialized models such as Qwen-Math-7B.

3. **More Robust Positive Reward Signals**: The core mechanism of FR3E generates more positive reward signals by encouraging structured exploration around high-entropy states. Empirical analysis over multiple rollouts per prompt shows that a higher proportion of generated trajectories are fully correct ("All-Right"), while fewer are completely incorrect ("All-Wrong").

By combining uncertainty-driven exploration with sampled intermediate feedback, FR3E supports a more stable and data-efficient RL training process. Its value-model-free design encourages meaningful trajectory expansion without relying on brittle reward shaping or complex critics.

## 2 RELATED WORK

**Reinforcement Learning: Challenges and Techniques**

RL faces key challenges when applied to sparse-reward or long-horizon tasks. Simple exploration strategies like $\epsilon$-greedy and Upper Confidence Bound (UCB) Auer (2002); Auer et al. (2002) are often ineffective in large state spaces. Structured methods such as Go-Explore Ecoffet et al. (2019; 2021) improve performance by archiving and revisiting promising states, a principle that inspires our work. Its extension, Intelligent Go-Explore (IGE) Lu et al. (2025), leverages foundation models to guide exploration. To handle temporal abstraction, Hierarchical RL (HRL) decomposes tasks into subgoals via "options" Sutton et al. (1999), which underpins methods that assign intermediate rewards Setlur et al. (2024), such as Process Reward Models (PRMs) Luo et al. (2024); Lightman et al. (2023), to mitigate the credit assignment problem (CAP) Pignatelli et al. (2023). CALM Luo et al. (2024) further automates subgoal identification and auxiliary reward assignment, improving credit attribution and training efficiency.

**Reinforcement Learning for Large Language Models**

RL enables LLMs to move beyond imitation learning by optimizing generation with task-specific rewards Ouyang et al. (2022). While PPO Schulman et al. (2017) is widely used, its value network is costly and unstable in the vast, discrete action space of language. This motivates value-free alternatives such as GRPO Shao et al. (2024) and RLOO Ahmadian et al. (2024), which rely on trajectory-level comparisons, and S-GRPO Dai et al. (2025), which promotes conciseness via reward decay. To improve credit assignment, VinePPO Kazemnejad et al. (2024) uses Monte Carlo rollouts from heuristic intermediate states, and PRIME Cui et al. (2025b) derives implicit process rewards from final outcomes to cut labeling costs. Despite progress, adapting structured exploration strategies like Go-Explore to autoregressive LLMs remains open.

## 3 PRELIMINARY

### 3.1 MDP SETUP FOR AUTOREGRESSIVE GENERATION

Many language-based reasoning tasks can be naturally formulated as sequential decision-making problems, which are typically modeled as Markov Decision Processes (MDPs) (Sutton et al., 1998). An MDP is defined by the tuple $(\mathcal{S}, \mathcal{A}, P, r, \gamma)$, where:

- $\mathcal{S}$ denotes the state space,
- $\mathcal{A}$ denotes the action space,
- $P : \mathcal{S} \times \mathcal{A} \times \mathcal{S} \to [0, 1]$ is the transition dynamics,
- $r : \mathcal{S} \times \mathcal{A} \to \mathbb{R}$ is the reward function, and
- $\gamma \in (0, 1)$ is the discount factor.

The objective of RL is to learn a policy $\pi(a|s)$ that maximizes the expected discounted return:

$$\mathbb{E}_\pi \left[ \sum_{t=0}^{T} \gamma^t r(s_t, a_t) \right], \tag{1}$$

where $T$ is the horizon of the task.

In the context of large language models (LLMs) (Ranzato et al., 2016; Ouyang et al., 2022), the environment corresponds to the autoregressive generation process. Formally, we define the components as follows:

- **States ($\mathcal{S}$):** Each state $s_t = (x_0, \ldots, x_m, y_0, \ldots, y_t)$ consists of the input context $x_{0:m}$ and the sequence of previously generated tokens $y_{0:t}$.
- **Actions ($\mathcal{A}$):** An action $a_t = y_{t+1} \in \mathcal{V}$ corresponds to selecting the next token from the vocabulary $\mathcal{V}$.
- **Dynamics ($P$):** The environment transitions deterministically to the next state $s_{t+1} = (x_0, \ldots, x_m, y_0, \ldots, y_{t+1})$.
- **Reward ($r$):** Rewards are typically sparse, often provided only at the end of a complete rollout (Schulman et al., 2017), which limits intermediate feedback and poses challenges for effective exploration.

### 3.2 REJECTION SAMPLING

In advantage estimation, when a prompt consistently produces identical rewards (all correct with reward=1 or all incorrect with reward=0), the advantage becomes zero, leading to vanishing policy gradients and reduced sample efficiency. To address this, we adopt *rejection sampling* (Yu et al., 2025; Zhang et al., 2025) during the *entropy finding* phase: for each prompt, multiple rollouts are generated and their binary rewards computed; if all rewards are identical, the prompt is deemed degenerate and removed from the batch. This ensures exploration focuses on diverse and informative trajectories rather than uninformative degenerate cases.

### 3.3 CLIP-HIGHER

Standard PPO constrains policy updates with symmetric clipping bounds (e.g., $[0.8, 1.2]$) to ensure stability, but this also suppresses the probability growth of low-confidence tokens, leading to *entropy collapse*. To alleviate this, we adopt the *Clip-Higher* mechanism (Yue et al., 2025), which introduces asymmetric clipping to allow larger probability increases for underexplored reasoning paths while still restricting excessive decreases. With $\epsilon_{\text{low}} = 0.22$ and $\epsilon_{\text{high}} = 0.28$, the objective becomes:

$$\mathcal{L}_{\text{PPO}}(\theta) = -\frac{1}{\sum_{i=1}^{G} |o_i|} \sum_{i=1}^{G} \sum_{t=1}^{|o_i|} \min \left( r_{i,t}(\theta) \hat{A}_{i,t}, \ \text{clip}(r_{i,t}(\theta), 1 - \epsilon_{\text{low}}, 1 + \epsilon_{\text{high}}) \hat{A}_{i,t} \right) \tag{2}$$

This asymmetric objective preserves policy diversity and improves exploration efficiency.

### 3.4 STRUCTURED EXPLORATION

Our method adapts the "First Return, Then Explore" paradigm from Go-Explore to LLM reasoning, providing a principled alternative to inefficient random exploration. In the **"First Return"** phase, we generate promising trajectories; in the **"Entropy-Eliciting Explore"** phase, we identify high-uncertainty states via local minima in token log-probabilities and initiate partial rollouts from these points. This structured process efficiently uncovers alternative reasoning paths, producing dense and informative feedback that strengthens the model's ability to refine solutions.

## 4 FR3E: FIRST RETURN, ENTROPY-ELICITING EXPLORE

Effective exploration in LLM reinforcement learning faces two primary obstacles: the loss of promising search trajectories and the inability to return to productive reasoning paths. Our approach, **FR3E**, addresses these challenges through a structured framework that explicitly separates the discovery of high-potential reasoning points from their systematic exploration.

The core idea is to decompose the reinforcement learning process into two complementary phases:

- **First Return**: Identify key positions along base trajectories where model uncertainty is highest — potential forks in the reasoning space.
- **Entropy-Eliciting Explore**: From these identified states, perform diverse rollouts to sample alternative solution paths while maintaining semantic coherence.

This structured paradigm enhances traditional exploration-exploitation trade-offs by incorporating entropy-based signals directly into both path selection and policy update mechanisms.

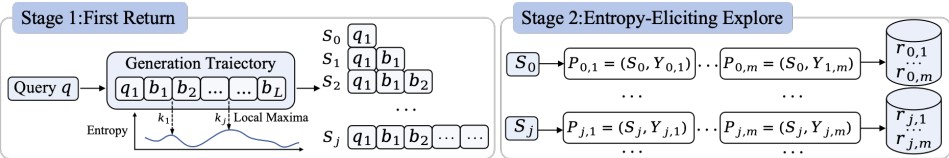

Figure 1: Overview of the FR3E framework. **stage 1: first return** begins with base trajectory generation from query $q$, followed by token-wise entropy computation to identify high-uncertainty positions. these positions serve as segmentation points for constructing intermediate semantic states $S_j$. **stage 2: entropy-eliciting explore** launches multiple rollouts from each state $S_j$, evaluates the reward for each, and computes empirical values $V(S_j)$ to guide adaptive policy updates. this two-stage design encourages diverse yet structured exploration based on model uncertainty signals.

### 4.1 FIRST RETURN: BLOCK DEFINITION VIA UNCERTAINTY SIGNALS

#### 4.1.1 BASE TRAJECTORY GENERATION

Given a training query $q$, we first generate a base reasoning trajectory using the policy $\pi_\theta$:

$$P_{\text{base}} = (q, t_1, t_2, \ldots, t_L) \tag{3}$$

where each $t_i$ denotes a generated token in the response sequence. This trajectory represents the model's step-by-step reasoning process from the initial query to the final output.

#### 4.1.2 ENTROPY COMPUTATION

To identify positions in the trajectory that exhibit high uncertainty — and thus are suitable for exploration — we compute the token-wise entropy at each position $k$. Specifically, let:

$$\pi_\theta(v \mid q, t_{<k}) \tag{4}$$

denote the softmax-normalized probability distribution over the vocabulary at step $k$, conditioned on the query $q$ and previously generated tokens $t_{<k}$.

We define the entropy of the policy at position $k$ as:

$$H_k = -\sum_{v \in \mathcal{V}} \pi_\theta(v \mid q, t_{<k}) \log \pi_\theta(v \mid q, t_{<k}) \tag{5}$$

where $\mathcal{V}$ is the model's vocabulary set.

Higher values of $H_k$ indicate greater uncertainty in the model's decision-making at that position. These high-entropy positions represent promising candidates for initiating structured exploration.

### 4.1.3 ENTROPY-SENSITIVE POSITION SELECTION

Given the full base trajectory $P_{\text{base}}$, we identify positions with maximal uncertainty by selecting the top-$K$ tokens with highest entropy values. Specifically, we extract a set of entropy-sensitive positions:

$$\mathcal{K} = \{k_1, \ldots, k_K\} \tag{6}$$

which correspond to the most uncertain decision points along the reasoning path.

**Top-$K$ Entropy Selection**: Select the indices corresponding to the $K$ largest entropy values across the entire trajectory (Wang et al., 2025):

$$\mathcal{K} = \text{TopK}\left(\{H_k\}_{k=1}^{L}\right) \tag{7}$$

This strategy ensures that we focus on the most uncertain reasoning steps globally, enabling us to identify natural forks in the model's reasoning space for structured exploration.

### 4.1.4 SEMANTIC BLOCK & STATE CONSTRUCTION

As shown in Figure 2, frequent tokens with high average entropy can be identified and used as key segmentation points. This figure is adapted from prior work (Wang et al., 2025), which analyzes token-level uncertainty across model trajectories. These entropy-sensitive positions serve as natural breakpoints for segmenting the trajectory into semantically meaningful reasoning blocks. Specifically, we define the block structure based on these key points:

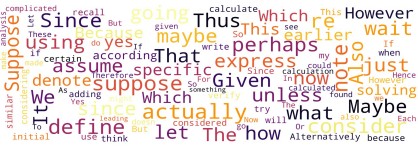

Figure 2: Frequent tokens with the highest average entropy.

$$P_{\text{base}} = B_1 \cup B_2 \cup \cdots \cup B_{K+1} \tag{8}$$

where each block $B_n$ contains the subsequence of tokens between two consecutive entropy-sensitive positions:

$$B_n = (t_{k_{n-1}+1}, \ldots, t_{k_n}), \quad \text{with } k_0 = 0, \; k_{K+1} = L \tag{9}$$

From the previously identified entropy-sensitive positions $\mathcal{K}$, we derive middle reasoning states:

$$S_j = (q, B_1, B_2, \ldots, B_j) \tag{10}$$

representing the full reasoning state up to and including block $B_j$, just prior to the entropy-sensitive position $k_j$.

This entropy-based segmentation provides structure for localized policy refinement and targeted credit propagation. It enables fine-grained control over which segments of the trajectory receive higher attention during optimization. By grounding exploration in model-intrinsic uncertainty signals, FR3E facilitates robust discovery of high-reward reasoning paths, particularly in tasks with sparse rewards and complex action spaces.

### 4.2 ENTROPY-ELICITING EXPLORE: DIVERSIFIED PATH SAMPLING FROM ANCHORS

For each intermediate state $S_j$, we reset the context and generate $M$ diverse reasoning rollouts:

$$\mathcal{Y}_j := \{Y_{j,m}\}_{m=1}^{M}, \quad Y_{j,m} \sim \pi_\theta(\cdot \mid S_j) \tag{11}$$

Each rollout $Y_{j,m}$ consists of a sequence of generated tokens extending from $S_j$. We define its reward based on whether the generated answer is correct:

$$r_{j,m} = \begin{cases} 1 & \text{if } Y_{j,m} \text{ is correct,} \\ 0 & \text{otherwise.} \end{cases} \tag{12}$$

Using this reward definition, we estimate the empirical value of $S_j$ (Kazemnejad et al., 2024) as:

$$V(S_j) = \frac{1}{M} \sum_{m=1}^{M} r_{j,m} \tag{13}$$

This value quantifies the average performance gain achieved when transitioning from the prefix $S_j$ to its extensions. It serves as a signal for adaptive policy updates during the Entropy-Eliciting Explore.

Through this mechanism, the model effectively samples from local neighborhoods around high-uncertainty prefixes, promoting diversity while preserving semantic coherence with the base trajectory. Specifically, by evaluating the correctness of each rollout, we can identify promising directions for exploration that are likely to lead to correct answers, thereby enhancing both the efficiency and effectiveness of the search process.

### 4.3 ADAPTIVE ADVANTAGE MODULATION FOR STABLE LEARNING

To leverage the dense reward signals from Entropy-Eliciting Explore, we introduce an adaptive mechanism that dynamically scales the advantage function. Specifically, an advantage modulation factor $\alpha_j = \exp(-(V(S_j) - V(S_{j-1})))$ adjusts the learning signal based on the marginal improvement in value between consecutive states. When $V(S_j) > V(S_{j-1})$, $\alpha_j < 1$ downscales the signal to prevent premature convergence, while $V(S_j) \le V(S_{j-1})$ yields $\alpha_j \ge 1$, amplifying the signal to encourage exploration. This mechanism functions as a feedback controller, stabilizing learning and promoting reasoning diversity. Under simplifying assumptions, the modulated advantage averages to zero across a batch, ensuring an approximately unbiased gradient estimator; full proof details are provided in the appendix C.

## 5 EXPERIMENTS

### 5.1 SETUP

**Data Source:** We compose the training set from two sources to balance stability and challenge: low-difficulty signals from *DeepScaler* and high-difficulty samples (levels 3–5) from *SimpleRL*. The combined corpus supports both foundational and advanced reasoning (see Appendix B).

**Training:** We use the VeRL framework with batch size $512$, learning rate $1 \times 10^{-6}$, and clip range $[0.22, 0.28]$. Responses are capped at 16k tokens, with 16 rollouts per prompt. KL and entropy regularization are disabled (KL Coeff = 0, entropy loss = 0). The mini-batch size is 128. Because rejection sampling may yield partial batches, we accumulate samples until a full batch is formed before each update.

**Benchmark:** We evaluate on GSM8K, Math500, Minerva Math, Gaokao2023en, OlympiadBench, College Math, and AIME24. Greedy decoding is used for all benchmarks except AIME24, which is assessed with *avg@32* (averaging 32 sampled rollouts per problem).

### 5.2 MAIN RESULTS

We compare FR3E and GRPO++ across Qwen2.5 variants (Qwen et al., 2025). As shown in Figure 3, all models exhibit an early entropy drop, but FR3E consistently maintains higher entropy than GRPO++, indicating stronger exploration and delayed convergence. This leads to improved AIME24 performance: FR3E yields steady gains on Qwen2.5-7B and -32B, whereas GRPO++ is less stable and occasionally fluctuates. In contrast, Qwen2.5-Math-7B saturates entropy earlier and shows weaker gains, suggesting that strong domain-specific priors constrain the benefits of RL fine-tuning. Finally, FR3E produces more stable advantage estimates (Appendix D.3), aligning with theory and supporting its overall training stability.

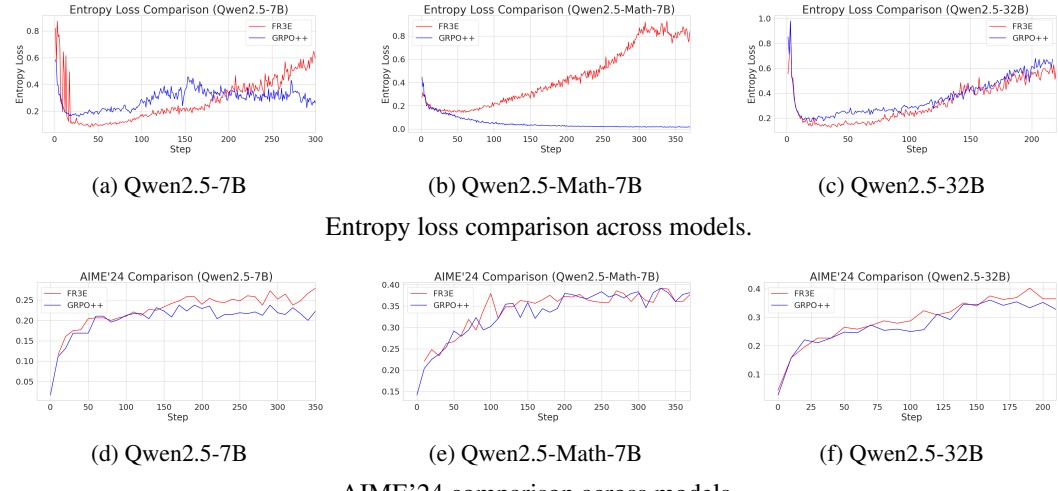

(a) Qwen2.5-7B      (b) Qwen2.5-Math-7B      (c) Qwen2.5-32B

Entropy loss comparison across models.

(d) Qwen2.5-7B      (e) Qwen2.5-Math-7B      (f) Qwen2.5-32B

AIME'24 comparison across models.

Figure 3: Combined results: [Top] Entropy loss comparison across models; [Bottom] AIME'24 comparison across models.

Table 1 summarizes results for FR3E vs. GRPO++ across eight math-reasoning benchmarks and three backbones, revealing three patterns. First, gains grow with capacity: on **Qwen2.5-32B** FR3E attains the highest average accuracy (65.6%), outperforming GRPO++ by +3.1%, with pronounced improvements on AIME24 (+6.1%) and AMC23 (+5.0%), indicating that DAG-based rollouts benefit from the larger reasoning space. Second, on general-purpose backbones (Qwen2.5-7B/32B), FR3E consistently surpasses GRPO++; on the domain-specialized **Qwen2.5-Math-7B**, the average advantage narrows to +1.8% (e.g., AIME24: 39.1% for both; GSM8K/Math500: +0.6%), suggesting strong priors reduce the incremental benefit of exploration-driven RL. Third, the largest relative gains occur on long-chain benchmarks—**Olympiadbench** (+3.0–3.5%), **AMC23** (+5.0–7.5%)—while shallow tasks (e.g., GSM8K) exhibit minor gains (+0.1–1.6%), showing FR3E is most effective for deep, multi-step reasoning. Overall, FR3E scales with model size, yields its strongest gains on general-purpose backbones, and is most advantageous on deep-reasoning benchmarks, while still offering modest, consistent improvements on specialized models.

In addition, FR3E produces more stable advantage estimates (see Appendix D.3), consistent with theoretical expectations.

## 5.3 DISCUSSION

### 5.3.1 TRAINING DYNAMICS OF QWEN-MATH-7B

As noted in the Main Results, the RL performance of Qwen2.5-Math-7B exhibits distinct patterns compared to other models, motivating a closer examination of its training dynamics. We observe several notable characteristics during training:

Table 1: Multi-benchmark comparison (accuracy %)

| Benchmark | Qwen2.5-Math-7B | | Qwen2.5-7B | | Qwen2.5-32B | |
| | FR3E | GRPO++ | FR3E | GRPO++ | FR3E | GRPO++ |
|---|---|---|---|---|---|---|
| AIME24 | 39.1 (+0.0%) | 39.1 | 25.2 (+2.5%) | 22.7 | 40.2 (+6.1%) | 34.1 |
| GSM8k | 91.3 (+0.1%) | 91.2 | 92.8 (+1.6%) | 91.2 | 96.1 (+0.3%) | 95.8 |
| Math500 | 82.2 (+0.6%) | 81.6 | 79.0 (+1.2%) | 77.8 | 87.4 (+2.2%) | 85.2 |
| Minerva Math | 40.8 (+2.6%) | 38.2 | 39.0 (+3.7%) | 35.3 | 45.6 (+2.6%) | 43.0 |
| Gaokao2023en | 67.8 (+2.6%) | 65.2 | 67.3 (+3.4%) | 63.9 | 75.3 (+3.9%) | 71.4 |
| Olympiadbench | 46.5 (+3.5%) | 43.0 | 42.1 (+3.3%) | 38.8 | 51.7 (+3.0%) | 48.7 |
| College Math | 47.4 (-0.1%) | 47.5 | 45.1 (+0.0%) | 45.1 | 48.3 (+1.3%) | 47.0 |
| AMC23 | 67.5 (+5.0%) | 62.5 | 67.5 (+7.5%) | 60.0 | 80.0 (+5.0%) | 75.0 |
| Avg | 60.3 (+1.8%) | 58.5 | 57.3 (+3.0%) | 54.3 | 65.6 (+3.1%) | 62.5 |

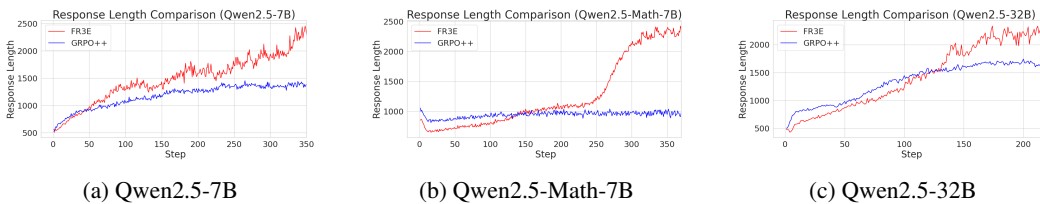


(a) Qwen2.5-7B      (b) Qwen2.5-Math-7B      (c) Qwen2.5-32B


Figure 4: Response length comparison across different models

**Early saturation in response length:** The model quickly reaches the maximum sequence length limit, which may constrain the development of longer reasoning paths.

**Overstable entropy trends:** Compared to general-purpose models, Qwen2.5-Math-7B exhibits larger entropy fluctuations and earlier drops in entropy loss, suggesting reduced exploration capacity and a tendency toward entropy collapse during training. When comparing training algorithms, Figure 4b shows that GRPO++ keeps response length relatively flat, whereas FR3E steadily increases it, indicating that its trajectory-level reward shaping encourages the development of extended reasoning sequences. This pattern is mirrored in AIME24 (Figure 3e), where GRPO++ produces fluctuating and less stable performance, while FR3E shows a consistent upward trend. These observations highlight that Qwen2.5-Math-7B exhibits unique training dynamics under standard RL methods. Importantly, FR3E appears to better support structured reasoning and stabilize learning progress. Overall, the combined evidence from entropy trends, response lengths, and task performance suggests FR3E offers clear advantages for promoting exploration and exploitation in RL for LLM.

### 5.3.2 SOURCES OF GAINS IN FR3E

The improvements of FR3E can be attributed to exploration and consistency. By maintaining higher entropy throughout training, FR3E fosters more robust exploration and delays convergence, particularly in larger models. In parallel, its trajectory-level reward shaping reduces "all-wrong" rollouts while reinforcing consistent high-quality reasoning, thereby ensuring more stable updates. Together, these mechanisms explain the observed gains across benchmarks and model sizes.

**Higher Entropy Enables Healthier Exploration:** Entropy loss trends across different model sizes reveal the exploration-exploitation dynamics of FR3E versus GRPO++. On Qwen2.5-7B (Figure 3a), both algorithms initially show a sharp entropy decrease, indicating rapid policy certainty gain. As training progresses, FR3E maintains higher entropy than GRPO++, reflecting sustained exploration and avoidance of premature convergence; a similar pattern appears at a different scale (Figure 3b) and on Qwen2.5-32B (Figure 3c), where FR3E gradually increases entropy while GRPO++ converges quickly. For Qwen2.5-Math-7B, entropy stabilizes early at a low level, yet AIME24 scores remain comparable to models with higher entropy trajectories, suggesting that domain-specific priors and sequence length constraints can limit the impact of exploration. Overall, FR3E's ability to sustain higher entropy supports better exploration, improving reasoning diversity, accuracy, and stability, with the effect more pronounced in larger models.

**Entropy-Eliciting Explore, Improving Trajectory Consistency:** We evaluate exploration consistency at the trajectory level by assessing, for each prompt, whether all rollouts are entirely incorrect ("All-Wrong") or entirely correct ("All-Right"). As shown in Figure 5, two key patterns emerge:

*FR3E improves consistency over time.* It substantially increases All-Right trajectories while suppressing All-Wrong ones, indicating a shift toward stable, high-quality reasoning rather than sporadic successes. The widening gap between the All-Right and All-Wrong curves suggests that trajectory reweighting accumulates reliable positive signals, yielding more stable and meaningful updates.

*Model-size and domain effects.* On Qwen2.5-Math-7B (Figure 5b, Figure 5e), despite early entropy convergence (Subsection 5.3.1), All-Right still rises moderately, implying that domain priors can bootstrap performance under limited exploration. Qwen2.5-7B (Figure 5a, Figure 5d) exhibits faster, more consistent gains, with a sharper increase in All-Right and a steeper decrease in All-Wrong. Qwen2.5-32B (Figure 5c, Figure 5f) achieve s the highest All-Right and lowest All-Wrong counts, underscoring the benefits of larger capacity for leveraging FR3E's exploration.

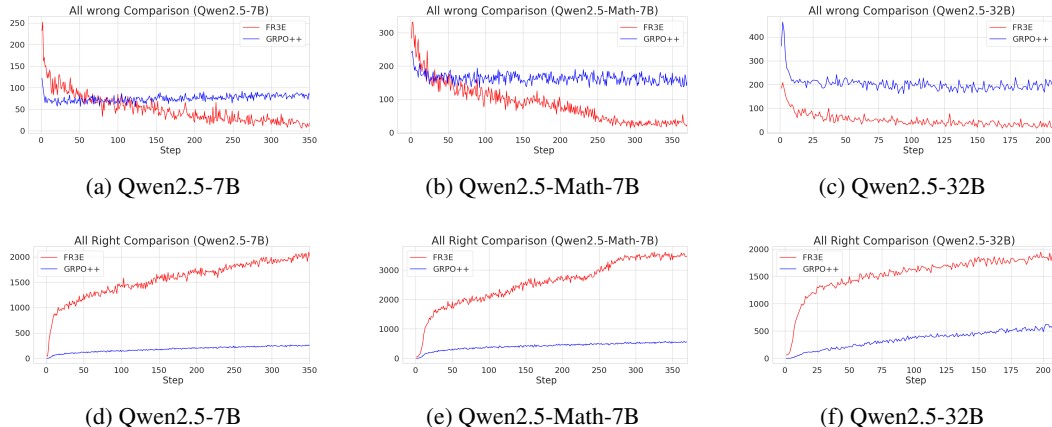

(a) Qwen2.5-7B      (b) Qwen2.5-Math-7B      (c) Qwen2.5-32B

(d) Qwen2.5-7B      (e) Qwen2.5-Math-7B      (f) Qwen2.5-32B

Figure 5: [TOP] Number of All-Wrong and [Bottom] All-Right trajectories during the entropy-eliciting explore phase.

### 5.3.3 TEST DISTRIBUTION ANALYSIS

To better understand the learning dynamics of FR3E during training and its impact on reasoning stability, we visualize rollout accuracy across all AIME problems for both FR3E and GRPO++. The heatmap for FR3E (Figure 6a) reveals a clear and progressive transition from low-accuracy regions to high-accuracy regions, indicating that the model steadily improves reasoning performance rather than relying on abrupt or sporadic gains (Li et al., 2025). Sharp peaks emerge where certain problems rapidly converge to consistently high accuracy, demonstrating that once a problem is mastered, FR3E retains this capability with minimal fluctuation. These observations reflect a stable and incremental learning pattern, where problem-solving skills are acquired gradually and internalized reliably — effectively "locking in" knowledge and ensuring high confidence in generated solutions while suppressing error propagation, which is especially critical for long-chain reasoning tasks. In contrast, the heatmap for GRPO++ (Figure 6b) shows greater variability, with some problems converging quickly while others remain at lower accuracy levels even after extensive training, highlighting potential instability in its learning process. Overall, FR3E's trajectory demonstrates both robustness and consistency, providing more dependable reasoning improvements than baseline.

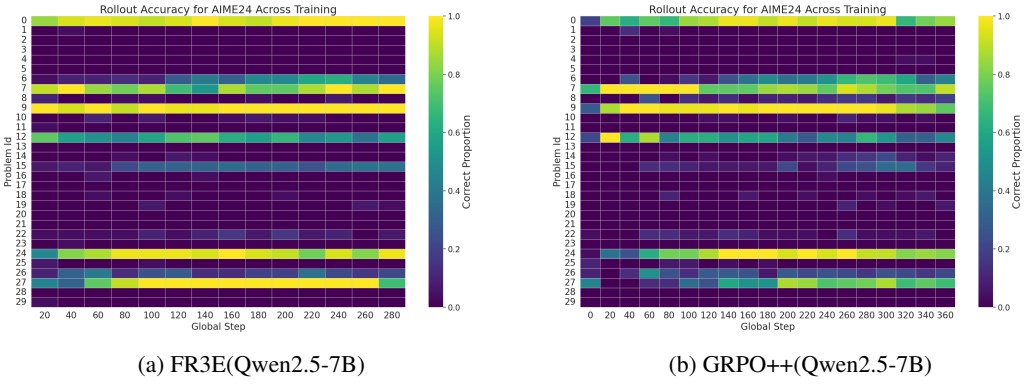

(a) FR3E(Qwen2.5-7B)          (b) GRPO++(Qwen2.5-7B)

Figure 6: Rollout accuracy heatmap comparison (Qwen2.5-7B)

## 6 CONCLUSION

We introduce **FR3E** (First Return, Entropy-Eliciting Explore), a framework to improve exploration in RL for LLMs. By detecting high-uncertainty tokens via entropy and launching targeted rollouts, FR3E gathers intermediate feedback and uses an adaptive advantage to stabilize updates. It yields more stable training, longer and more coherent reasoning chains, and higher solution accuracy, offering a structured path to more reliable exploration.

ETHICS STATEMENT

The authors of this work have read and commit to adhering to the ICLR Code of Ethics. Our research on **FR3E** (First Return, Entropy-Eliciting Explore) focuses on reinforcement learning methods for improving reasoning in large language models (LLMs). We address ethical considerations as follows:

**Human Subjects.** This work does not involve new human subject data collection. For evaluation, we rely exclusively on publicly available benchmarks (e.g., AIME24, GSM8K, MATH) that are widely used in the research community. No personal, identifiable, or sensitive information is included in these datasets.

**Data Sourcing and Copyright.** All datasets used are publicly available and licensed for research purposes. We ensure that our usage complies with dataset licenses and respect original authors' copyrights. Our contributions are methodological, and we do not introduce new text data beyond what is already released under open terms.

**Bias and Societal Impact.** As with any work involving LLMs, our method may inherit biases from the underlying pretrained models and datasets. While FR3E improves reasoning stability, it does not explicitly address fairness or bias mitigation. We caution practitioners that deploying models trained with FR3E in downstream applications should involve careful auditing for potential biases and unintended societal impacts.

REPRODUCIBILITY STATEMENT

We have taken concrete steps to ensure the reproducibility of our findings:

**Code and Resources.** The full implementation of FR3E, including training scripts, evaluation pipelines, and logging utilities, is released at the anonymized repository: `https://anonymous.4open.science/r/FR3E-59ED`.

**Methodology.** The FR3E algorithm, including entropy-based trajectory sampling, advantage scaling, and rollout scheduling, is described in detail in 4. Additional pseudocode and hyperparameter configurations are provided in C.

**Experimental Setup.** Complete details of the training configurations, including model checkpoints (Qwen2.5-7B, Qwen2.5-Math-7B, Qwen2.5-32B), hyperparameters, and computing infrastructure (number of GPUs, batch sizes, learning rates), are reported in 5.1.

Together, these resources are intended to make our work fully reproducible by independent researchers.

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

## A  USE OF LARGE LANGUAGE MODELS

We employ LLMs solely for language polishing and clarification, such as refining sentences for readability and grammatical correctness. No LLMs were involved in the development of research concepts, the design of the FR3E algorithm, the implementation of experiments, or the interpretation of results. Additionally, LLMs were occasionally used to assist in generating test prompts or to validate code logic in edge-case scenarios, but all primary research decisions and analyses were performed by the authors.

## B  DATA SETTING CLAIM

In our preliminary experiments, we explored alternative data settings. Specifically, we conducted a comparative study on the **Qwen2.5 7B math base** model. The results indicated that models trained using data from the DAPO dataset yielded inferior performance compared to our current data configuration, which combines the **DeepScaler** and **SimpleRL** datasets.

Consequently, we adopted the current setting for our main experiments. For the sake of experimental consistency and inertia, this data setting was maintained across all subsequent model training processes. However, it is important to note that this conclusion is based on our specific observations with the Qwen2.5 7B math base model. The findings may not be directly transferable or hold true for other base models, such as the **Qwen2.5 32B base**, which might exhibit different behaviors and data requirements.

## C  PROOF OF ADAPTIVE ADVANTAGE MODULATION PROPERTIES

**Definition.**  For each state $S_j$, we define the modulation factor as

$$\alpha_j = \frac{1}{\exp(V(S_j) - V(S_{j-1}))}, \tag{14}$$

and the modulated advantage for a trajectory $P_{j,m}$ as

$$A'(S_j, P_{j,m}) = \alpha_j \cdot A(S_j, P_{j,m}), \tag{15}$$

where $A(S_j, P_{j,m}) = r_{j,m} - V(S_j)$ is the standard advantage.

**Unbiasedness under idealized conditions.**  Assume all trajectories within a block $j$ have identical length $L_j$. Then the batch-averaged modulated advantage is

$$\bar{A}' = \frac{1}{|\mathcal{T}|} \sum_j \alpha_j \sum_{(j,m) \in O_j} (r_{j,m} - V(S_j)), \tag{16}$$

where $|\mathcal{T}|$ is the total number of tokens in the batch.

Since each trajectory contributes exactly $L_j$ tokens, the inner sum simplifies to

$$\sum_{(j,m) \in O_j} (r_{j,m} - V(S_j)) = L_j \sum_{m=1}^{M} (r_{j,m} - V(S_j)). \tag{17}$$

By the definition of $V(S_j)$ as the empirical mean value, this sum equals zero. Therefore, each term in the outer summation vanishes, leading to

$$\bar{A}' = 0. \tag{18}$$

**Practical implications.** In practice, due to minor variations in trajectory lengths and stochasticity in value estimation, $\bar{A}'$ may deviate slightly from zero. Nonetheless, the modulation mechanism yields an approximately unbiased gradient estimator while effectively incorporating dense feedback obtained through structured exploration.

# D ADDITIONAL RESULTS AND ANALYSIS

## D.1 ENTROPY DYNAMICS

Figure 3 illustrates entropy loss across different model sizes. For Qwen2.5-7B and 32B, FR3E sustains higher entropy throughout training, while GRPO++ converges rapidly to lower values. This suggests that FR3E maintains a more favorable exploration–exploitation balance, avoiding premature convergence that may hinder the discovery of novel reasoning trajectories. In contrast, Qwen2.5-Math-7B exhibits an earlier plateau in entropy, likely reflecting the influence of its domain-specific pretraining. The specialized inductive biases of math-oriented models may reduce the marginal benefits of reinforcement-driven exploration.

## D.2 RESPONSE LENGTH AND STABILITY

As shown in Figure 4, FR3E enables longer and more consistent reasoning chains compared to GRPO++, which maintains relatively flat response lengths. Longer chains are generally associated with more detailed and verifiable reasoning processes, especially in multi-step tasks. FR3E's ability to sustain such trajectories suggests that it encourages structured reasoning rather than shortcutting to premature answers. Figure 3 further highlights that FR3E exhibits a smoother upward trajectory on AIME24, while GRPO++ fluctuates more significantly. This stability is critical for domains where single errors in reasoning chains can propagate and degrade final accuracy.

## D.3 ADVANTAGE TRENDS

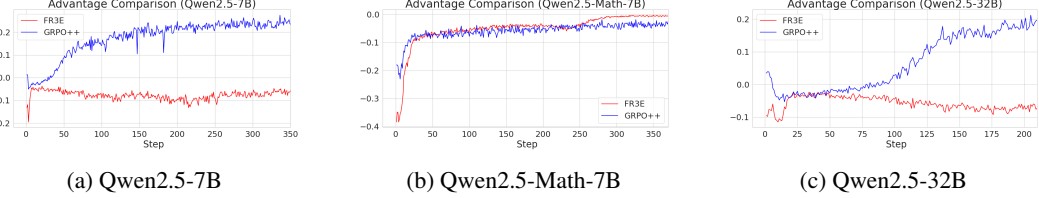

| (a) Qwen2.5-7B | (b) Qwen2.5-Math-7B | (c) Qwen2.5-32B |

Figure 7: Advantage comparison across different models

Figure 7 compares the moving averages of advantage values for FR3E and GRPO++. FR3E closely aligns with the theoretical expectation $\mathbb{E}_{a\sim\pi(\cdot|s)}[A(s,a)] = 0$, reflecting reduced variance and more stable training dynamics. This alignment indicates that FR3E effectively mitigates distributional mismatch between the rollout policy and the training policy, thereby lowering the risk of unstable updates. In contrast, GRPO++ shows larger oscillations around zero, suggesting noisier gradients and less predictable learning behavior.

## D.4 RESOURCE CONSIDERATIONS

We note that GRPO++ employs group rollouts (default size 4), while FR3E operates on a DAG-based trajectory structure. As a result, inference cost alignment is not exact; FR3E typically incurs slightly higher inference costs. However, these costs remain sub-linear with respect to rollout depth, since intermediate states can be reused across partial rollouts. The performance gains reported in Section 5.2 suggest that this additional cost is justified: modestly higher compute yields disproportionate improvements in stability and generalization, particularly on large backbones.

