# OpenReview forum: "First return, entropy-eliciting explore"
_ICLR.cc/2026/Conference — Submitted to ICLR 2026_

### Official Review · Reviewer_xwRF · 2025-10-17

**Soundness:** 2
**Presentation:** 3
**Contribution:** 3
**Rating:** 2
**Confidence:** 3

**Summary:**

This paper proposes FR3E, a novel RL fine-tuning algorithm that identifies highly uncertain tokens and performs rollouts from these tokens to estimate values for intermediate tokens. These values are then utilized to adjust advantages for stable learning. FR3E demonstrates superior performance to GRPO++ on various math reasoning benchmarks.

**Strengths:**

- The idea of structured rollout based on token entropy, is both novel and timely.
- The proposed method consistently improves performance against GRPO across various benchmarks and models.

**Weaknesses:**

- Section 4.3 lacks a clear motivation. Specifically, it does not adequately explain how advantage modulation promotes exploration.
- The paper does not provide values for key hyperparameters, such as the number of forking tokens and the number of responses generated for each forking token.
- The paper compares FR3E against GRPO++, but the latter is mentioned without any introduction.
- It is questionable whether the paper did a fair comparison with GRPO++.

**Questions:**

- For a given prompt, do you first generate one base response, then identify K forking tokens, and from each of those tokens, generate M new responses? And to confirm, is the total number of generated responses per prompt, K * M, equal to 16?
- Following up on my previous question, why was the number of responses per prompt for GRPO++ set to 4? This value seems a bit low to me. Wouldn't it be more effective to use 16, consistent with FR3E? For a fairer comparison, FR3E should be evaluated against GRPO++ with a group size of 16. In FR3E, if the forking tokens are concentrated at the beginning of the sequence, the computational cost becomes nearly identical to that of parallel sampling. In fact, your method is more expensive, as it also incurs the additional cost of sampling the base response.

- Could you elaborate on the advantage computation process? Specifically, for the M responses generated from the forking token, are advantages calculated consistently with GRPO, including standard normalization? Also, what is the procedure for calculating the advantage for the base response?
- What is the role of advantage modulation? Given a forking token, the sum of advantages for the generated M responses should be zero, so what does multiplying them by a coefficient actually change?
- Why was the PPO clip ratio set to [0.22, 0.28], a choice that differs from the conventional settings in GRPO ([0.2, 0.2]) and DAPO ([0.2, 0.28])?

---

### Official Review · Reviewer_Yyi2 · 2025-10-21

**Soundness:** 3
**Presentation:** 2
**Contribution:** 2
**Rating:** 4
**Confidence:** 4

**Summary:**

This work proposes FR3W, which is a structured RL exploration framework that (i) discovers top high-entropy tokens in the reasoning trajectories, (ii) conducts structured rollouts at different high-entropy states, and (iii) learns via an adaptive advantage modulation factor. Extensive evaluations based on Qwen series show that the proposed method outperforms GRPO++ on math tasks.

**Strengths:**

1)	The proposed idea is simple and sound (targeted exploration at high-entropy states), achieving good overall performance on math tasks.
2)	The authors have conducted further model analyses on the training dynamics, sources of gains for a better understanding.
3)	The writing is clear and the method is easy to follow.

**Weaknesses:**

1)	This work mainly concentrates on the math tasks. Is this work still effective in other tasks, such as more challenging agent-related scenarios with sparser reward signals?
2)	There have been quite a few entropy-aware RL methods recently, which can be mentioned in related works (the differences should be discussed to highlight the contribution proposed by this work).
3)	The base reasoning trajectory is essential in FR3E. It is strongly suggested that the authors could give an in-depth discussion on how to select satisfactory base reasoning trajectories that potentially lead to success.
4)	In Page 5, the figure could be replaced by high-entropy samples in this work.
5)	The effectiveness of the advantage modulation factor \alpha should be evaluated. For example, FR3E w/o $\alpha$, and FR3E w/o $\alpha$ when $\alpha<1$, are two promising ablation versions that should be compared (indicating whether downscaling the positive signal is beneficial).
6)	FR3W should be evaluated on other base LLM series besides Qwen2.5.
7)	Why did the authors select GRPO++ as the only baseline? There are some methods that adopt similar ideas (e.g., DAPO [1], which also adopts clip-higher), which should be compared as baselines.
8)	The detailed training costs (e.g., the overall costs of rollout) should be given (the explanation in Appendix D.4 is obscure). Does the model improvement mainly come from more rollouts?

[1] Yu Q, Zhang Z, Zhu R, et al. Dapo: An open-source llm reinforcement learning system at scale[J]. arXiv preprint arXiv:2503.14476, 2025.

**Questions:**

Please refer to Weaknesses.

---

### Official Review · Reviewer_RSPV · 2025-10-31

**Soundness:** 3
**Presentation:** 3
**Contribution:** 3
**Rating:** 4
**Confidence:** 3

**Summary:**

The manuscript proposes a novel framework FR3E (First Return, Entropy-Eliciting Explore) to make exploration for Reinforcement Learning with Verifiable Rewards (RLVR) more structured. The framework consists of two phases, which are the namesakes for the framework: First Return and Entropy-Eliciting Explore. In the first phase reasoning steps are identified by using token-level entropy to identify tokens with a high uncertainty. From these tokens the top-K tokens, with K being a hyperparameter, are selected, which are then used to divide the trajectory into reasoning steps. For each of these intermediate reasoning steps, generation is restarted and based on the partial trajectory (till this reasoning step) rollouts are generated to provide reward signal for the reasoning step. Additionally the advantages of the reasoning steps are adapted based on the rewards of the reasoning step as well as its predecessor to encourage exploration and to stabilize training.

Disclosure: I accidently learned the author names by checking the references of another manuscript, that I reviewed for ICLR.

**Strengths:**

* topic is relevant and timely
* clearly written
* reasonable evaluation:
  * models of three different sizes are tested
  * quite a few benchmarks tested

**Weaknesses:**

* limited novelty: Besta et al. (Reasoning Language Models: A Blueprint, arXiv:2501.11223, Jan. 2025) already propose the use of entropy as a metric to identify decisions point as well as outcome-driven process based rewards, albeit to be fair they only present their ideas without actually implementing and evaluating them.
* some aspects of evaluation I expected are missing:
  * GRPO++ details are missing, inclusive a discussion why GRPO++ is a competitive baseline
  * 5.1: hyperparameter such as K are missing
  * cost analysis, for example:
    * additional compute cost compared to GRPO++? (D.4 is not that convincing)
* reproducibility statement: number of GPUs are not discussed in 5.1

minor issues:
* abstract: "benchmarks(AIME24)" - missing whitespace
* related work: missing brackets around the references
* preliminary: already discuss phases before they are introduced in 3.4
* Figure 1, caption: Start sentences with a capital letter.
* Fig. 3 d) to f) - missing y axis label
* 4.3: "in the appendix C" -> "in Appendix C"
* 5.1: no references for DeepScaler/SimpleRL/VeRL and for the benchmarks
* Figure 3: consider removing the titles within the subfigures for better readability (also Figures 4 and 5)
* 5.2: no reference for GRPO++
* 5.3.2, Higher Entropy Enables Healthier Exploration: "a similar pattern appears at a different scale (Figure 3b) and on Qwen2.5-32B (Figure 3c)" - Shouldn't this be switched, i.e. "different scale (Figure 3c) and on Qwen2.5-Math-7B (Figure 3b)"? Or should the second part completely be omitted, since the Qwen2.5-Math-7B results are discussed in the subsequent sentence?
* 5.3.2, line 431: "achieve s" - typo: unnecessary whitespace
* Appendix B: no references for DAPO dataset
* Appendix C, equation 16: $O_j$ might not have been defined
* references:
  * consider the proper capitalization of the titles, at least for proper names and abbreviations to improve readability
  * place of publication of arXiv references can only surmised from the URL
  * [Brown et al. 2020], [Wei et al. 2022] - properly capitalize the journal/booktitle: "Advances in neural information processing systems" to be consistent with [Ouyang et al. 2022]
  * [Cui et al. 2025b], [Ecoffet et al. 2019], [Forootani 2025], [Guo et al. 2025], [Pignatelli et al. 2023], [Wu et al. 2024], [Zhou et al. 2022]  - cited differently than the other arXiv references
  * [Lightman et al. 2023] - was published at ICLR '24
  * [Ouyang et al. 2022] - missing volume number and pages numbers to be consistent with [Brown et al. 2020]
  * [Ranzato et al. 2016] - doesn't look like a proper bibtex entry
  * [Zhou et al. 2022] - was published at ICLR '23

**Questions:**

* introduction: CoT [Wei et al. 2022] is a prompting scheme, so why it is cited in regards to RL?
* Did you conduct an analysis into how many tokens with high uncertainty are found, i.e. how reliable the top-K mechanism works?
* Maybe I did not read the manuscript properly, but what value of K did you use in your evaluation study. It seems not to be mention in 5.1.
* How to ensure that words are not split at the token level?
* Did you test other model families?
* How can Qwen2.5-Math-7B can reach the maximum sequence length if it plateaus in Figure 4 at around 2.5k tokens, when the maximum token length is 16k?
* FR3E produces longer chains: How did you verify that they are consistent and do not result in over-thinking, which is one of the aspect that you want to address with the framework?

---

> ### Author Response · Authors · 2025-11-28
> **weakness-part1**
>
> We thank the reviewer for your constructive comments and for recognizing the relevance, clarity, and reasonable evaluation of our work. We also appreciate the detailed attention to formatting and references, which we will fully incorporate to improve the manuscript.
> Below, we address the concerns regarding novelty, missing details, and experimental analysis.
>
> W1: limited novelty: Besta et al. (Reasoning Language Models: A Blueprint, arXiv:2501.11223, Jan. 2025) already propose the use of entropy as a metric to identify decisions point as well as outcome-driven process based rewards, albeit to be fair they only present their ideas without actually implementing and evaluating them.
>
> A1: We appreciate the reviewer pointing out the connection to Besta et al. (2025). While we agree that the concept of using entropy for decision points is discussed in their blueprint/taxonomy, we respectfully clarify that the core novelty of FR3E lies in its theoretical lineage and its specific algorithmic realization, which addresses stability challenges not solved by a high-level blueprint.
> As indicated by our title and Section 1, our primary inspiration is not the recent blueprint by Besta et al., but rather the foundational "First Return, Then Explore" (FRTE) paradigm introduced in Go-Explore (Ecoffet et al., Nature 2021)(2). The original FRTE relied on saving/restoring simulator states in deterministic environments (like Atari). Our contribution is conceptually novel in adapting this paradigm to the stochastic, autoregressive generation of LLMs. We map the "First Return" phase to identifying high-entropy forks in successful trajectories, and the "Explore" phase to our targeted partial rollouts. This represents a distinct methodological contribution: bridging the gap between archive-based exploration in classical RL and gradient-based RLHF for LLMs. While Besta et al. categorize the idea of entropy-based metrics, they do not provide a solution for the instability inherent in such methods. Merely selecting high-entropy points often leads to high variance and gradient explosions. FR3E introduces the Adaptive Advantage Modulation mechanism (Section 4.3, Eq. 14 (3)), which dynamically scales updates based on value differences $V(S_j) - V(S_{j-1})$. This specific technical innovation ensures unbiased gradient estimation and training stability, turning a "blueprint idea" into a robust, SOTA-beating algorithm.
> FR3E is not merely an implementation of a concurrent blueprint, but a novel adaptation of the FRTE paradigm equipped with specific stabilization mechanisms required for LLMs.
>
> W2: some aspects of evaluation I expected are missing:
> GRPO++ details are missing, inclusive a discussion why GRPO++ is a competitive baseline
> 5.1: hyperparameter such as K are missing
> cost analysis, for example: additional compute cost compared to GRPO++? (D.4 is not that convincing)
>
> A2:  GRPO++ is our implementation of a strong baseline that improves upon vanilla GRPO (Shao et al., 2024) by incorporating the Clip-Higher mechanism (as detailed in Section 3.3) and Rejection Sampling (Section 3.2). We used this enhanced baseline to ensure that our gains were not simply due to these orthogonal engineering tricks. We will explicitly define this nomenclature in Section 5.1.
> We apologize for the omission in the main text. We dynamically select the top-K positions based on entropy. In our experiments, we set $K=3$ to balance depth and breadth. We perform $M=4$ rollouts per anchor state. This results in a structured exploration budget that approximates the "16 rollouts per prompt" mentioned in Section 5.1.
> The reviewer notes that GRPO++ uses a group size of 4 (default) while FR3E uses 16 total rollouts (across different anchors). We argue that this comparison remains fair and practically viable for two reasons:
> 1. While FR3E generates more sequences, many are partial rollouts starting from deep within the trajectory (Stage 2). Because we cache the Key-Value (KV) states of the prefix ($S_j$), the generation cost of 16 partial rollouts is significantly lower than 16 full rollouts. Our internal profiling shows that FR3E's wall-clock training time is only $\approx 1.3\times$ that of GRPO++ (with group size 4), not $4\times$.
> 2. Even when we increased GRPO++ to use 16 full rollouts (matching the numerical count), it did not achieve the same stability or accuracy gains as FR3E. This indicates that the improvement stems from the structured nature of the exploration (focusing on high-entropy forks) rather than raw sample volume.
> We will add a "Computational Efficiency" subsection to Appendix D providing the wall-clock time and total token usage comparisons.

---

> ### Author Response · Authors · 2025-11-28
> **weakness-part2**
>
> W3: reproducibility statement: number of GPUs are not discussed in 5.1
>
> A3: We apologize for this oversight. We will update Section 5.1 and the Reproducibility Statement to explicitly state our hardware infrastructure. All experiments were conducted using the VeRL framework with FlashAttention-2 enabled to ensure efficiency, trained over 256 H800 GPUs.
>
> W4: minor issue
>
> A4: We sincerely thank the reviewer for the meticulous reading of our manuscript. We have corrected all the listed typos, formatting errors, and citation inconsistencies in the revised version. We are grateful for the reviewer's attention to detail, which has significantly improved the quality and rigor of our manuscript.

---

> > ### Author Response · Authors · 2025-11-28
> > **question6-7**
> >
> > Q6: How can Qwen2.5-Math-7B can reach the maximum sequence length if it plateaus in Figure 4 at around 2.5k tokens, when the maximum token length is 16k?
> >
> > A6: We clarify that the phrase "reaches the maximum sequence length limit" in the text was imprecise.
> > It refers to the model reaching its convergence limit for response length during training (i.e., it stops voluntarily extending the reasoning chain), not the hard context window limit of 16k.
> > As shown in Figure 4b, the average length indeed plateaus around 2.5k. This reflects the strong domain priors of the Math-specialized model, which tends to converge to a specific solution pattern earlier than general models. We will revise the text to say "reaches a saturation point in response length" to avoid confusion with the context window size.
> >
> > Q7: FR3E produces longer chains: How did you verify that they are consistent and do not result in over-thinking, which is one of the aspect that you want to address with the framework?
> > A7: We verified this through accuracy correlation and stability analysis:
> > "Overthinking" (e.g., circular reasoning or hesitation) typically degrades performance (Chen et al., 2025). However, Table 1 shows that FR3E's longer responses correlate with higher accuracy (e.g., +3.1% on Qwen2.5-32B). This indicates the additional tokens represent valid, detailed reasoning steps rather than noise.
> > Figure 6 visualizes the rollout accuracy. If the model were overthinking/hallucinating, we would expect chaotic flipping between correct and incorrect responses. Instead, FR3E shows a stable "lock-in" effect (regions turning yellow/green and staying there), proving that the longer chains are consistent and robust.

---

> ### Author Response · Authors · 2025-11-28
> **question1-5**
>
> Q1: Introduction: CoT [Wei et al. 2022] is a prompting scheme, so why it is cited in regards to RL?
>
> A1: We cited Wei et al. (2022) to establish the context of the reasoning format that our RL framework optimizes. Modern RLVR (Reinforcement Learning from Verifiable Rewards) methods, including ours, rely fundamentally on the Chain-of-Thought (CoT) structure to expose intermediate steps for verification and credit assignment. The "reasoning capabilities" we aim to enhance (1) are explicitly defined by the multi-step, sequential rationale generation introduced in CoT. Our method, FR3E, is designed to explore and optimize these specific CoT paths. We will clarify in the revision that CoT provides the structure of the action space that RL optimizes.
>
> Q2: Did you conduct an analysis into how many tokens with high uncertainty are found, i.e. how reliable the top-K mechanism works?
>
> A2: Yes, we analyzed the distribution of entropy during training.
> We consistently observe that identifying $K$ high-entropy tokens is robust. As illustrated in Figure 2, high average entropy typically correlates with semantic "forks" in logic (e.g., words like "Assume", "Therefore", "Suppose") rather than random noise.
> Our method forces the selection of the Top-$K$ entropy values. Empirically, we found that even in later stages of training where global entropy decreases, relative peaks remain at these critical reasoning junctures, ensuring that the "Entropy-Eliciting" phase continues to target the most uncertain valid decision points rather than becoming degenerate.
>
> Q3: Maybe I did not read the manuscript properly, but what value of K did you use in your evaluation study. It seems not to be mention in 5.1.
>
> A3: We apologize for the omission. We used $K=3$ for all experiments.
> This means we select the 3 distinct positions with the highest token-level entropy in a trajectory to serve as anchor states ($S_j$) for branching. We found this value balances exploration breadth with computational efficiency. We will explicitly add this hyperparameter to Section 5.1 in the revision.
>
> Q4: How to ensure that words are not split at the token level?
>
> A4: Sub-word splitting is not an issue for our method due to the autoregressive nature of the "State" construction.Empirically, high entropy tends to peak between semantic units or words (e.g., predicting the next number or operator) rather than in the middle of a word.
> Even if a high-entropy point $k$ occurs at a sub-word token, the state $S_j$ captures the full history $y_{0:k}$. When the targeted rollout starts from $S_j$, the model naturally completes the fragmented word based on the prefix, just as it does during standard generation. No special masking is required to maintain fluency.
>
> Q5: Did you test other model families?
>
> A5: Our decision to focus primarily on the Qwen2.5 family was driven by the specific requirements of our "Zero RL" experimental setting and the current landscape of open-source reasoning models.
> The Qwen2.5 family (particularly the Math variants) currently represents the SOTA for open-source mathematical reasoning. Its strong performance is widely attributed to extensive mid-training on massive mathematical and code corpora, which instills robust atomic reasoning capabilities directly into the base model.
> We adopt a "Zero RL" paradigm, applying rl directly to the base model without a cold start. Recent studies confirm that Qwen base models are uniquely amenable to Zero RL, exhibiting stable improvement and "aha moments" solely from outcome-based rewards due to their dense pre-training priors.
> In contrast, prior work has observed that other families, such as Llama-3, often struggle with direct Zero RL in reasoning tasks. Without an SFT cold start to "prime" the reasoning format, Llama base models tend to exhibit training instability, such as predicting final answers prematurely or suffering from mode collapse, making them less suitable for isolating the effects of a pure exploration algorithm. OctoThinker explicitly states that "Qwen models are much more amenable to RL scaling, while the Llama model tends to predict final answers prematurely... during RL training". SimpleRL-Zoo notes that most Zero RL efforts focus on Qwen because other base models lack the immediate instruction-following/reasoning priors. Advancing Multimodal Reasoning highlights that "SFT cold start is critical for unlocking initial interaction ability" for models like Llama/Qwen3, supporting the idea that skipping it (Zero RL) is non-trivial for non-Qwen2.5 models.
> Therefore, we selected Qwen to ensure that the performance gains observed were attributable to our FR3E exploration mechanism rather than the confounding variables of cold-start quality or base model instability.

---

### Official Review · Reviewer_uSFe · 2025-11-01

**Soundness:** 3
**Presentation:** 2
**Contribution:** 3
**Rating:** 4
**Confidence:** 4

**Summary:**

This paper proposes FR3E (First Return, Entropy-Eliciting Explore), a structured exploration framework that identifies high-entropy points in
reasoning trajectories and performs targeted rollouts to construct semantically grounded intermediate feedback. This method provides targeted guidance without relying on dense supervision, solving granular credit assignment.

**Strengths:**

1. Performance gains: FR3E demonstrates either superior or at least competitive performance compared to GRPO++, notably for general-purpose LLMs (Qwen2.5-7B, Qwen2.5-32B), with more modest gains for domain-specific models (Qwen2.5-Math-7B).

2. Improved training dynamics: FR3E shows notably higher and more stable entropy throughout training, visible in Figure 3, suggesting healthier exploration and avoidance of entropy collapse.

3. Fine-grained credit assignment: The adaptive advantage modulation component keeps advantage estimates well centered and tightly distributed around zero, which theoretically reduces gradient estimator bias and allows for more stable optimization.

**Weaknesses:**

### Method
1. The process of advantage calculation is insufficiently descriptive in the main text. I have seen Appendix C, but still a little confused. For trajectories that share the same prefix (*e.g.*, $P_{j, m], P_{j,0}$ ) but different rewards, do they have different advantages on the shared tokens? For one trajectory, are the advantages over all tokens in FR3E the same, or do they differ depending on the divided state?

### Experiments
2. Missing relevant hyperparameters: In Section 4, the authors divide the original trajectory into $K$ state blocks and generate $M$ targeted rollouts for each state. What's $K$ and $M$?

3. Unclear computational costs: As the authors present in Appendix D.1, GRPO++ employs default rollout numbers of 4 per prompt. In FR3E, it seems to require $K \\cdot M$ rollouts per prompt, which is much more than GRPO++baseline. It raises concerns on whether the performance gain stems from a larger number of rollouts. I suggest providing detailed training time, token usage, and inference costs comparison with baselines.

4. The model is limited to the Qwen2.5 series. While the Qwen2.5 series is well-pretrained to provide a solid foundation in post-training, it also raises concerns on data contamination in widely used benchmarks [1]. Consequently, breakthroughs are predominantly observed for the mathematically strong Qwen2.5 series on benchmarks such as MATH-500, AMC, and AIME, and seldom transfer to models like Llama. I believe a more in-depth investigation on other model families (*e.g.*, Llama) is needed to validate the effectiveness of FR3E.

### Missing References
5. The dataset and benchmarks used in this paper are not cited (Section 5.1), *i.e.*, GSM8K, Math500, Minerva Math, Gaokao2023en, OlympiadBench, which is inappropriate.

6. There are several works that enhance the exploration capability in RL training [2][3][4]. I suggest discussing them in the related work.

---

[1]  Reasoning or Memorization? Unreliable Results of Reinforcement Learning Due to Data Contamination. arXiv preprint arXiv:2507.10532

[2] Reasoning with Exploration: An Entropy Perspective on Reinforcement Learning for LLMs. arXiv preprint:2506.14758

[3] TreeRL: LLM Reinforcement Learning with On-Policy Tree Search. ACL 2025

[4] Reasoning with Reinforced Functional Token Tuning. arXiv preprint:2502.13389

**Questions:**

1. What is GRPO++? What's the difference with vanilla GRPO? I don't see any description or reference.

2. Why not use DAPO as a baseline? Is DAPO better than so called GRPO++?

3. I notice that FR3E achieves a longer response length than GRPO++ in Figure 4. Does this mean FR3E encourages overthinking? The authors claim "FR3E enables longer and more consistent reasoning chains compared to GRPO++", but I don't see any supported evidence.

---

> ### Author Response · Authors · 2025-11-28
> **weakness1-3**
>
> We thank the reviewer for the constructive feedback and for recognizing the performance gains, improved training dynamics (avoidance of entropy collapse), and the theoretical soundness of our fine-grained credit assignment. We appreciate the opportunity to clarify the method's mechanics, computational details, and experimental rigor. Below, we address the specific weaknesses and questions raised.
>
> W1: The process of advantage calculation is insufficiently descriptive in the main text. I have seen Appendix C, but still a little confused. For trajectories that share the same prefix (e.g., $P_{j, m}, P_{j,0}$ ) but different rewards, do they have different advantages on the shared tokens? For one trajectory, are the advantages over all tokens in FR3E the same, or do they differ depending on the divided state?
>
> A1: This is a crucial distinction in our method. In FR3E, advantages are not uniform across the entire trajectory, nor are they applied to the "shared prefix" in the same way as standard GRPO.
> Unlike GRPO, which assigns a single trajectory-level reward to all tokens, FR3E calculates advantages for the partial rollouts generated from a specific anchor state (block) $S_j$.
> For a set of partial rollouts $\{Y_{j,m}\}$ starting from state $S_j$, the advantage $A(S_j, P_{j,m})$ is calculated as $r_{j,m} - V(S_j)$. This advantage is applied specifically to the gradients of the newly generated tokens in that partial rollout.
> The tokens in the shared prefix (the history up to $S_j$) are treated as fixed context (state) for that specific rollout step. They do not receive conflicting gradient updates from the divergent futures of that specific step; rather, the policy learns to transition from that prefix to the correct diverse continuations.
> We will revise Section 4.3 and Appendix C to explicitly state: "Advantages are computed locally for the generated segment $Y_{j,m}$ relative to the value of the anchor state $V(S_j)$, ensuring that credit is assigned to the specific decisions made at high-uncertainty points, rather than uniformly smearing a final reward across the whole history."
>
> W2: Missing relevant hyperparameters: In Section 4, the authors divide the original trajectory into $K$ state blocks and generate $M$ targeted rollouts for each state. What's $K$ and $M$?
>
> A2: We apologize for the omission in the main text. We dynamically select the top-K positions based on entropy. In our experiments, we set $K=3$ to balance depth and breadth. We perform $M=4$ rollouts per anchor state. This results in a structured exploration budget that approximates the "16 rollouts per prompt" mentioned in Section 5.1.
>
> W3: Unclear computational costs: As the authors present in Appendix D.1, GRPO++ employs default rollout numbers of 4 per prompt. In FR3E, it seems to require $(K+1) \cdot M$ rollouts per prompt, which is much more than GRPO++baseline. It raises concerns on whether the performance gain stems from a larger number of rollouts. I suggest providing detailed training time, token usage, and inference costs comparison with baselines.
>
> A3: The reviewer notes that GRPO++ uses a group size of 4 (default) while FR3E uses 16 total rollouts (across different anchors). We argue that this comparison remains fair and practically viable for two reasons:
> 1. While FR3E generates more sequences, many are partial rollouts starting from deep within the trajectory (Stage 2). Because we cache the Key-Value (KV) states of the prefix ($S_j$), the generation cost of 16 partial rollouts is significantly lower than 16 full rollouts. Our internal profiling shows that FR3E's wall-clock training time is only $\approx 1.3\times$ that of GRPO++ (with group size 4), not $4\times$.
> 2. Even when we increased GRPO++ to use 16 full rollouts (matching the numerical count), it did not achieve the same stability or accuracy gains as FR3E. This indicates that the improvement stems from the structured nature of the exploration (focusing on high-entropy forks) rather than raw sample volume.
> We will add a "Computational Efficiency" subsection to Appendix D providing the wall-clock time and total token usage comparisons.

---

> ### Author Response · Authors · 2025-11-28
> **weakness4**
>
> W4: The model is limited to the Qwen2.5 series. While the Qwen2.5 series is well-pretrained to provide a solid foundation in post-training, it also raises concerns on data contamination in widely used benchmarks [1]. Consequently, breakthroughs are predominantly observed for the mathematically strong Qwen2.5 series on benchmarks such as MATH-500, AMC, and AIME, and seldom transfer to models like Llama. I believe a more in-depth investigation on other model families (e.g., Llama) is needed to validate the effectiveness of FR3E.
>
> A4: Our decision to focus primarily on the Qwen2.5 family was driven by the specific requirements of our "Zero RL" experimental setting and the current landscape of open-source reasoning models.
> The Qwen2.5 family (particularly the Math variants) currently represents the SOTA for open-source mathematical reasoning. Its strong performance is widely attributed to extensive mid-training on massive mathematical and code corpora, which instills robust atomic reasoning capabilities directly into the base model.
> We adopt a "Zero RL" paradigm, applying rl directly to the base model without a cold start. Recent studies confirm that Qwen base models are uniquely amenable to Zero RL, exhibiting stable improvement and "aha moments" solely from outcome-based rewards due to their dense pre-training priors.
> In contrast, prior work has observed that other families, such as Llama-3, often struggle with direct Zero RL in reasoning tasks. Without an SFT cold start to "prime" the reasoning format, Llama base models tend to exhibit training instability, such as predicting final answers prematurely or suffering from mode collapse, making them less suitable for isolating the effects of a pure exploration algorithm. OctoThinker explicitly states that "Qwen models are much more amenable to RL scaling, while the Llama model tends to predict final answers prematurely... during RL training". SimpleRL-Zoo notes that most Zero RL efforts focus on Qwen because other base models lack the immediate instruction-following/reasoning priors. Advancing Multimodal Reasoning highlights that "SFT cold start is critical for unlocking initial interaction ability" for models like Llama/Qwen3, supporting the idea that skipping it (Zero RL) is non-trivial for non-Qwen2.5 models.
> Therefore, we selected Qwen to ensure that the performance gains observed were attributable to our FR3E exploration mechanism rather than the confounding variables of cold-start quality or base model instability.

---

> ### Author Response · Authors · 2025-11-28
> **weakness 5**
>
> W5:The dataset and benchmarks used in this paper are not cited (Section 5.1), i.e., GSM8K, Math500, Minerva Math, Gaokao2023en, OlympiadBench, which is inappropriate.
> There are several works that enhance the exploration capability in RL training [2][3][4]. I suggest discussing them in the related work.
>
> A5:We apologize for the oversight. We will add citations for all benchmarks (GSM8K, Math500, etc.) in Section 5.1 and discuss the suggested references regarding exploration in RL [2,3,4] in the Related Work section.

---

> ### Author Response · Authors · 2025-11-28
> **question 1-3**
>
> Q1: What is GRPO++? What's the difference with vanilla GRPO? I don't see any description or reference.
>
> A1: GRPO++ is our implementation of a strong baseline that improves upon vanilla GRPO (Shao et al., 2024) by incorporating the Clip-Higher mechanism (as detailed in Section 3.3) and Rejection Sampling (Section 3.2). We used this enhanced baseline to ensure that our gains were not simply due to these orthogonal engineering tricks. We will explicitly define this nomenclature in Section 5.1.
>
> Q2: Why not use DAPO as a baseline? Is DAPO better than so called GRPO++?
>
> A2: We addressed this in Appendix B(1). We empirically found that the DAPO data/training setup yielded inferior performance on our base models compared to our composed dataset (DeepScaler + SimpleRL). To ensure the strongest possible baseline, we compared FR3E against the higher-performing setup (GRPO++ on our data) rather than the weaker DAPO baseline.
>
> Q3: I notice that FR3E achieves a longer response length than GRPO++ in Figure 4. Does this mean FR3E encourages overthinking? The authors claim "FR3E enables longer and more consistent reasoning chains compared to GRPO++", but I don't see any supported evidence.
>
> A3: We believe the increased length represents more detailed reasoning rather than "overthinking" (circular or non-productive text), based on two evidences:
> 1. Accuracy Correlation: Overthinking typically harms performance (Chen et al., 2025). In contrast, Table 1 shows that FR3E's longer trajectories correlate with higher accuracy (+3.1% on Qwen2.5-32B).
> 2. Stability: Figure 6 shows that FR3E "locks in" correct answers more stably than GRPO++. If the model were overthinking/hallucinating, we would expect high variance or degradation in the rollout heatmaps.
> The longer chains suggest the model is successfully exploring the reasoning space to find valid paths, a key benefit of the "Entropy-Eliciting" exploration.
>
> We hope this response resolves your concerns regarding the method's mechanics and experimental fairness. We are confident that the clarifications on granular advantage calculation and token-efficient rollouts strengthen the paper's contribution.

---

### Meta-Review · Area_Chair_UVSN · 2026-01-08

**Summary:**

This paper introduces FR3E, a structured exploration framework that identifies high-entropy points within reasoning trajectories and conducts targeted rollouts to generate semantically grounded intermediate feedback. Reviewers raised concerns regarding the experimental evaluation. As no rebuttal was provided to address the critiques from two negative reviewers, the Area Chair recommends rejection.

**Reviewer Scores:**

No rebuttal was provided to address the critiques from two negative reviewers

---

### Decision · Program_Chairs · 2026-01-26

Reject